# The Role of One-Carbon Metabolism in Healthy Brain Aging

**DOI:** 10.3390/nu15183891

**Published:** 2023-09-07

**Authors:** Sapna Virdi, Abbey M. McKee, Manogna Nuthi, Nafisa M. Jadavji

**Affiliations:** 1Department of Biomedical Sciences, Midwestern University, Glendale, AZ 85308, USA; svirdi89@midwestern.edu (S.V.); amckee92@midwestern.edu (A.M.M.); manogna.nuthi@midwestern.edu (M.N.); 2College of Osteopathic Medicine, Midwestern University, Glendale, AZ 85308, USA; 3College of Veterinary Medicine, Midwestern University, Glendale, AZ 85308, USA; 4Department of Child Health, College of Medicine Phoenix, University of Arizona, Phoenix, AZ 85308, USA; 5Department of Neuroscience, Carleton University, Ottawa, ON K1S 5B6, Canada

**Keywords:** neurodegeneration, healthy aging, one-carbon metabolism, folic acid, vitamin B12, choline

## Abstract

Aging results in more health challenges, including neurodegeneration. Healthy aging is possible through nutrition as well as other lifestyle changes. One-carbon (1C) metabolism is a key metabolic network that integrates nutritional signals with several processes in the human body. Dietary supplementation of 1C components, such as folic acid, vitamin B12, and choline are reported to have beneficial effects on normal and diseased brain function. The aim of this review is to summarize the current clinical studies investigating dietary supplementation of 1C, specifically folic acid, choline, and vitamin B12, and its effects on healthy aging. Preclinical studies using model systems have been included to discuss supplementation mechanisms of action. This article will also discuss future steps to consider for supplementation. Dietary supplementation of folic acid, vitamin B12, or choline has positive effects on normal and diseased brain function. Considerations for dietary supplementation to promote healthy aging include using precision medicine for individualized plans, avoiding over-supplementation, and combining therapies.

## 1. Introduction

The world’s population is aging, and the number of age-related diseases is on the rise [1]. Healthy aging is possible through nutrition as well as other lifestyle changes [2]. One-carbon (1C) metabolism is a key metabolic network that integrates nutritional signals with biosynthesis, redox homeostasis, and epigenetics. One-carbon metabolism plays an essential role in the regulation of cell proliferation, stress resistance, and embryo development [3,4]. In the brain, 1C plays an important role in methylation, lipid metabolism, DNA repair, and purine synthesis (Figure 1). 

B-vitamins such as vitamin B9 (folic acid) and vitamin B12, as well as the nutrient choline, play important roles in 1C (Figure 1). For example, they are all involved in the methylation of a non-protein amino acid called homocysteine. Increased levels of homocysteine indicate deficiencies, genetic or nutritional, within the 1C pathway. Some clinical studies have demonstrated that increased levels of homocysteine have been associated with negative outcomes for brain health, such as Alzheimer’s disease [5], stroke [6], vascular dementia [7], as well as neural tube defects [8] in developing babies. However, the link between elevated levels of homocysteine and neurodegeneration are not clear [9].

Deficiencies in 1C can arise through either genetic changes or reduced intake from diet. As individuals age, there are changes in physiology which can lead to deficiencies in 1C components. For example, changes in stomach acid pH and reduced levels of intrinsic factor led to less vitamin B12 absorption from the diet resulting in a deficiency [10]. Reduced levels of vitamin B12 have been linked to increased risk of stroke and worse outcomes [11]. 

In terms of healthy aging, dietary supplementation of 1C may be a prudent step to take prior to the onset of neurodegeneration [9,12]. The aim of this review is to summarize the current clinical studies investigating dietary supplementation of 1C, specifically folic acid, choline, and vitamin B12, and its effects on healthy aging. We have also included a preclinical study section that provides details on the mechanisms through which supplementation may be changing brain function to promote healthy aging. This article will also discuss future steps to consider for the dietary supplementation of folic acid, choline, and vitamin B12. 

## 2. Folic Acid

### 2.1. Functional Role of Folic Acid

Homocysteine plays an important role in methylation reactions in the body and is dependent on folic acid, as well as vitamins B6 and B12, for synthesis [13]. Folic acid deficiency results in elevated homocysteine levels, and hyperhomocysteinemia has been identified as a risk factor for various diseases, such as vascular disease. In patients with hypertension, the role of folic acid supplementation on SAH (S-adenosylhomocysteine) levels was measured [14]. This study also investigated the effect of the 5,10-methylenetetrahydrofolate reductase (MTHFR) C677T gene polymorphism on SAH levels. MTHFR is another enzyme that plays a role in folate metabolism. The patients recruited for this study were aged 45–75 years old with a history of primary hypertension and were grouped based on their MTHFR C677T polymorphism (CC, CT, or TT). The treatment groups included a daily oral dose of 10 mg enalapril with either 0.4, 0.6, 0.8, 1.2, 1.6, 2.0, or 2.4 mg of folic acid. Treatment groups administered 10 mg of enalapril with 0.4–2.0 mg of folic acid did not have altered S-adenosylhomocysteine (SAH) levels, but supplementation with 2.4 mg increased SAH levels. Patients with the MTHFR C677T genotype CT and TT supplemented with 2.4 mg of folic acid had increased SAH levels. The CC genotype did not show an increase in SAH levels with the supplementation of 2.4 mg of folic acid. These findings indicated that the MTHFR genotypes affected by folic acid supplementation were both CT and TT. Additionally, higher levels of folic acid supplementation (2.4 mg) resulted in increased homocysteine levels like what is reported during folate deficiency. This suggests that higher levels of folic acid supplementation could be harmful, which was confirmed by two other studies [15,16]. Low and high serum folate concentrations were associated with increased risk of mortality related to cardiovascular disease. 

Telomere attrition or shortening is a key finding in many age-related disorders. The role of folic acid in telomere shortening was investigated, noting that folic acid had been implicated in astrocyte (glial cell) apoptosis and in aging disorders, but its role in telomere attrition was unknown [17]. Four-month-old male mice were fed diets with varying folic acid concentrations, including 0.1 (folic acid deficient diet), 2.0 (folic acid normal diet), 2.5 (low folic acid supplemented diet), and 3.0 mg (high folic acid supplemented) of folic acid per kg of diet for six months. The mice were euthanized when they were 10 months old, and astrocytes in the hippocampal and cerebral cortex tissue were analyzed, along with telomere length and telomerase activity. Folic acid supplementation prevented astrocyte apoptosis, telomere shortening and apoptosis, and degeneration in both the hippocampus and cortex. Telomerase activity was also shown to increase with folic acid supplementation, which is likely what prevented telomere attrition. 

### 2.2. Folic Acid’s Implications in Neurodegeneration

The role of 1C metabolism in cognitive health has been explored [18,19]. Altered 1C metabolism through dietary deficiencies in B-vitamins may have a role in cognitive decline resulting from changes in DNA methylation [18], increased homocysteine levels [19], and decreased levels of neurotransmitters, as well as nucleotides [20]. Inflammation has been implicated in cognitive impairment and conditions such as Alzheimer’s disease [21]. The link between inflammation and folate status is also being studied, specifically the role of folic acid supplementation in inflammation and cognition in patients with Alzheimer’s disease. The participants for this study were patients aged 40–90 years old with new diagnoses of mild to moderate Alzheimer’s disease who were being treated with Donepezil. The control group included patients taking just the Donepezil, and the treatment group included patients taking Donepezil with 1.25 mg of oral folic acid daily for six months with assessment every six weeks. Inflammation levels were determined through the measurement of IL-6 and TNFα mRNA. The effect of folic acid supplementation on present Alzheimer’s disease pathology was determined through measurements of serum Aβ (amyloid beta) 40, Aβ 42, Aβ 42/Aβ 40, APP (amyloid precursor protein) mRNA, PS1 (presenilin 1) mRNA, and PS2 mRNA. Cognition was studied through Mini-Mental State Examination (MMSE) score and Activities of Daily Living (ADL) score. Participants in the treatment group had higher levels in the MMSE compared to the control group indicating improved cognition with folate supplementation. However, levels of Aβ 40 decreased after folic acid supplementation leading to a higher Aβ 42/Aβ 40 ratio which is typically the case in familial Alzheimer’s disease. SAM (S-adenosylmethionine) increased post-treatment, and TNFα mRNA decreased indicating lower levels of inflammation. Prior studies have demonstrated a relationship between increasing SAM levels and decreasing TNFα. It was concluded that folic acid supplementation along with Donepezil treatment improved cognition, as well as inflammation in patients with Alzheimer’s disease. 

Another study investigated folic acid and vitamin B12 supplementation on inflammation and cognitive health in Alzheimer’s disease patients [22]. The patients for this study had a diagnosis of “stable” Alzheimer’s disease with a Montreal Cognitive Assessment score of less than 22 and were taking medication individually prescribed to them for their diagnosis. The treatment group was administered 1.2 mg of folic acid and 50 μg of vitamin B12 orally per day for six months, while the control or placebo group was administered the equivalent number of starch tablets resembling both folic acid and vitamin B12. Blood samples were analyzed for folate, vitamin B12, SAM, and SAH levels. There was also quantification of Aβ 40, Aβ 42, and the following inflammatory markers: IL2, IL6, IL10, MCP1, and TNF-α. Cognitive health was tested through neuropsychological testing including the Montreal Cognitive Assessment and the Alzheimer’s Disease Assessment Scale-Cognitive subscale. Supplementation of both folic acid and vitamin B12 resulted in higher total, naming, and orientation scores of the Montreal Cognitive Assessment. Scoring for attention improved for the Alzheimer’s Disease Assessment Scale-Cognitive subscale. The results of this neuropsychiatric testing indicated that the combined supplementation improved cognition in these patients. Combined supplementation also increased SAM and SAM/SAH levels and decreased homocysteine levels. Levels of inflammatory marker TNFα also decreased indicating a decrease in inflammation with combined folic acid and vitamin B12 supplementation. 

A similar study aimed to determine the effect of folic acid and vitamin B12 (separately and combined) on patients with mild cognitive impairment (MCI) [9]. Participants were divided into four groups, including the control group that was not administered treatment, folic acid alone with 400 μg of folic acid daily, vitamin B12 alone with 25 μg daily, and combined treatment with 400 μg of folic acid and 25 μg of vitamin B12 daily for 6 months. Blood analysis was completed for the quantification of inflammatory cytokines: IL2, IL6, IL10, TNFα, IFN-y, and MCP-1. Cognitive testing was completed through the Wechsler Adult Intelligence Scale-Revised (WAIS-RC). Findings revealed that combined treatment decreased inflammatory cytokines IL6, TNFα, and MCP-1 while improving cognitive testing (WAIS-RC) scores.

Another example of the positive effects of dietary supplementation with folic acid, vitamin B6, and vitamin B12 is that they reduce the risk of age-related macular degeneration (AMD), which is the leading cause of severe irreversible vision loss in the elderly [23]. The study supplemented female participants that had a preexisting cardiovascular disease with 2.5 mg/day, vitamin B6 50 mg/day, and vitamin B12 1.0 mg/day for 7.3 years; there were 55 cases of AMD in the treated group, and 82 in the placebo group. Other studies investigating 1C supplementation have also reported that dietary supplementation is beneficial for people at high risk of AMD [23,24]. 

### 2.3. Folic Acid Interactions with Vitamin B12

The effect of high serum folate was studied in elderly individuals with a history of diabetes and vitamin B12 deficiency [25]. Prior studies have shown that high folate supplementation can result in increased cognitive impairment [26]. There is also evidence that folate supplementation in individuals with vitamin B12 deficiency can result in cognitive impairment [27]. The goal of this study was to determine if there was a correlation between high levels of folate supplementation and cognitive impairment in individuals with vitamin B12 deficiency. The participants for this study were recruited from an established study of vitamin B12 supplementation in elderly Chinese individuals (average of 75 years old) with a history of diabetes. Serum methylmalonic acid and folate levels were assessed in these individuals, and those with high methylmalonic acid with a concentration greater than 0.3 μmol and high serum folate with a concentration greater than 31.4 nmol/L were chosen for this study. Magnetic resonance imaging was used to analyze brain structure in relation to methylmalonic and folate levels in these individuals. Imaging revealed atrophy of the gray matter in the right middle occipital gyrus, as well as the inferior frontal gyrus, in individuals with high folate concentrations and vitamin B12 deficiency. This led to the conclusion that high folate concentrations could be harmful to neuronal structure resulting in degeneration in the setting of vitamin B12 deficiency. 

## 3. Vitamin B12

Data suggest that as healthy humans age, both males and females exhibit a decrease in plasma concentrations of vitamin B12 as the body ages and metabolism changes [10]. Vitamin B12 deficiency has been linked to cognitive and memory impairment in some individuals, as shown with inverse correlations between methyl malonic acid (MMA) and global cognition and executive function tests [28,29,30,31]. Research also suggests that in addition to decreased levels of B12 in healthy individuals, the brain atrophies at a slow, steady rate, decreasing in volume over time, but this is exacerbated when deficient in vitamin B12 [29]. Very few studies deny that B12 supplementation shows significant reduction in total plasma homocysteine in patients, and multiple studies have demonstrated that B12 supplementation may somehow be linked to decreasing rates of brain atrophy and volume loss, primarily in patients with high concentrations of homocysteine at baseline.

In a study performed in 2020, 92 amnesic mild cognitive impairment (aMCI) patients were split into two groups, control (46 cases) or treatment (46 cases), to assess whether folate and vitamin B12 supplementation would decrease total serum homocysteine and improve cognitive function [32]. Both study groups received routine treatment, but the treatment group received an additional 5.0 mg/day folate + 500 µg × 3/day vitamin B12. The groups were assessed prior to treatment, and at 4, 12, and 24 weeks of treatment. As anticipated, participants in the treatment group demonstrated significant and steady increased concentrations of total serum folate and vitamin B12, with inversely related concentrations of homocysteine, at weeks 4, 12, and 24. This suggests that supplementation of folate and vitamin B12 sufficiently serves in decreasing total homocysteine. As intervention time increased, the Montreal Cognitive Assessment Scale score improved significantly by the 24th week compared to before treatment, and with respect to the control group at the same time point. These data show that, with respect to intervention duration, neurocognition function increased with decreased levels of total homocysteine. Additionally, the intervention group exhibited a significantly shorter P300 latency at 24 weeks compared to before treatment, intra-group latency at 12 weeks, and the control group at 24 weeks, despite a lack of significant change in amplitude for either group at all time points. P300 potential is an assessment of cognitive function by using the Oxford Multimedia Electromyography (EMG) system to assess response times to an electrical stimulus. This further supports that with decreased concentrations of total homocysteine at 24 weeks of treatment, the neurocognitive function of aMCI patients improved. It is important to keep in mind that the results of this study were based on a small sample of amnesic MCI patients.

Where the literature demonstrates the most inconsistency is regarding whether vitamin B12 supplementation also affects cognitive function, for example, episodic and spatial memory. Vitamin B12 supplementation in participants diagnosed with dementia and B12 deficiency, without folate deficiency, was responsible for significant increases in the Mini Mental State Exam (MMSE) scores, decreased hippocampus atrophy, decreased homocysteine, and increased vitamin B12 levels in plasma [33]. Another randomized, double-blinded study suggests that patients diagnosed with mild cognitive impairment (MCI), and especially in patients with hyperhomocysteinemia, demonstrate an impeded regression in neurocognition when supplemented with vitamin B12 [34]. However, other studies suggest that, despite vitamin B12 supplementation, cognitive and memory impairment persist despite biochemical restorations to healthy concentrations [35]. In patients that have a history of hypertension, anemia, or are healthy without MCI, several studies suggest the sole benefit of vitamin B12 supplementation is to decrease plasma homocysteine concentrations [36,37]. These studies suggest a lack of long-term benefits in immediate recall or attention, or cognitive function in general [38]. 

## 4. Choline

In healthy elderly populations, choline supplementation has an overall positive effect on cognition [39]. In a cross-sectional study, it was reported that choline supplementation had a neuroprotective effect on the elderly [40]. Specifically, in the 187.60 to 399.0 mg/day intake range, there was a significantly lower risk of cognitive impairment of 50%, as assessed by the CERAD (Consortium to Establish a Registry for Alzheimer’s Disease), AF (Animal Fluency), and DSST (Digital Symbol Substitution) tests of cognitive function [40]. Another study further supported this positive correlation by looking at the four neuropsychological factors of verbal memory, visual memory, verbal learning, and executive function along with white matter hyperintensity [39]. Higher choline intake was associated with better performance in the four factors, and there was a significant inverse relationship to white matter hyperintensity in a non-demented healthy population of elderly adults [39]. Another form of choline, known as citicoline or CDP-choline, was measured for its effects on elderly populations with age-associated memory impairment. This form of choline also showed significant improvements in overall memory, especially episodic memory, compared to the placebo group [41].

Contrary to the elderly populations, a double blind, placebo controlled cross-over experiment that assessed choline bitartrate supplementation showed no significant improvement in acute memory performance in young adults [42]. However, this study did not look at long-term effects for comparison, and the rapid turnaround from supplementation to memory testing may have limited the results. 

## 5. Preclinical Studies of Folic, Choline, and Vitamin B12 Supplementation Using Model Systems

In an effort to understand the mechanisms through which the supplementation of folic acid, vitamin B12, and choline impacts brain function, we have reviewed model system studies. These studies have an important role in our understanding of mechanisms. It is important to note that some of the preclinical studies we have included in this review article were not conducted in aged model systems. There are several challenges involved when using aged animals such as increased costs and translation to humans [43,44]. 

### 5.1. Brain Metabolism Is Affected by Folate and Vitamin B12 Status

As the brain ages, total volume decreases [45], as well as glucose metabolism [46] in the brain. Folic acid supplementation can prevent cognitive impairment, but its role in glucose metabolism is unknown. Researchers aimed to identify if folic acid status resulted in structural or metabolic changes in the brain [47]. Three-month-old male rats were administered diets containing either 0.1 (folic acid deficient), 2.0 (folic acid normal), 4.0 (low folic acid supplementation), or 8.0 mg (high folic acid supplementation) per kg diet for 22 months. Magnetic resonance imaging (MRI) and diffusion tensor imaging was completed to determine changes in brain structure. Folic acid supplementation reduced age-induced atrophy in the hippocampus. Brain positron emission tomography (PET) for glucose distribution in the brain revealed that folic acid supplementation resulted in increased glucose uptake in the brain. Behavioral testing was carried out using the Morris Water Maze test and the Open Field test. Testing revealed that folic acid supplementation altered age-related cognitive impairment. Overall, folic acid supplementation resulted in improved glucose brain metabolism and decreased age-related structural changes in the hippocampus and age-related cognitive decline.

*S*-adenosylmethionine (SAM) plays a direct role in methylation reactions, and folate deficiency results in a decline in SAM [48]. SAM also controls neurotransmitter levels, and thus a decline in folate levels may result in altered cognition, as is seen in Alzheimer’s disease through altered acetylcholine levels. A 5,10-methylene tetrahydrofolate reductase (MTHFR) deficiency has also been identified to reduce choline levels, which ultimately results in lower acetylcholine. Folic acid deficiency and its role on cognitive health and behavior through SAM and acetylcholine was studied in adult (9- to 12-month-old) and aged (2.0- to 2.5-year-old) mice [49]. In this study, both a dietary folic acid deficiency and oxidative stress-induced folate deficiency were included. There were varying mice strains used, including “normal” adult and aged mice, *Mthfr*^+/+^ and *Mthfr*^−/−^ mice, and mice with murine apolipoprotein (ApoE) knocked out, as ApoE4 is a gene predictive for the onset of Alzheimer’s disease, and these mice additionally either expressed an empty vector, ApoE2, ApoE3, or ApoE4. The mice were administered two different diets: a diet deplete of folate and vitamin E with iron (to induce oxidative stress), or a diet supplemented with 4mg/kg folic acid and vitamin E for one month. A separate group of mice on the deficient diet were also administered a supplementation of 100 mg/kg SAM for one month. Following this, cognitive testing was carried out via the standard Y maze test, and behavior (aggression) was also observed in the mice at that time. Folate-depleted “normal” and *Mthfr*^+/+^ mice had cognitive impairment per Y maze testing, but mice expressing ApoE2 and ApoE3 did not. *Mthfr*^+/−^ mice on a “complete” diet with folate supplementation also showed cognitive impairment compared to the *Mthfr*^+/+^ mice. However, on the folate-depleted diet, both groups showed increasing cognitive impairment. Mice expressing ApoE3, ApoE4, or MTHFR on a folate-deficient diet showed aggression. The mice were killed, and the cortical and hippocampal areas were assayed to quantify levels of acetylcholine and SAM. The findings revealed that folate deficiency resulted in declined levels of SAM in all mice. Supplementation of SAM in the folate-deficient mice resulted in repleted acetylcholine levels in the mice and improved aggression, as well as cognitive impairment. These results led to the conclusion that SAM supplementation can play a role in the repletion of acetylcholine even in the setting of dietary or stress-induced folate deficiency while also improving cognitive impairment and altered behavior. 

Altered 1C metabolism through folate deficiency and its role in neuroprogenitor cell proliferation has not been studied extensively [50]. One study carried out in adult mice revealed that folate deficiency reduced the proliferation of cells in the hippocampal dentate gyrus (in vivo) and reduced the proliferation of embryonic neuroprogenitor cells (in vitro). For one part of the study, mice embryo trunks were dissected and incubated in folate-deficient solution or solution containing methotrexate, which inhibited folate metabolism. Cell counts were performed to quantify neuroprogenitor cells, which revealed that folate deficiency led to decreased proliferation of these cells. For the in vivo study, one-month-old mice were maintained on a standard or folate-deficient diet for 3.5 months and then sacrificed. Similar to the in vitro findings, there was decreased proliferation of progenitor cells in the dentate gyrus, thus producing the conclusion that deficient one-carbon metabolism through decreased folate and elevated homocysteine altered neuroprogenitor cell proliferation. 

Using aged rats with hyperhomocysteinemia, researchers assessed folate and B12 supplementation that reduced levels of homocysteinemia-induced tau deposition in the hippocampus, and the reversal of statistically significant spatial memory deficits without any impairment to their learning ability or alterations to the tau biochemical signaling [51]. These results suggest the need to perform clinical tests on whether folic acid and B12 supplementation can reverse, slow, or stop the progression of hyperhomocysteinemia and prevent the deposition of tau aggregates. 

### 5.2. Mechanisms of Choline Supplementation

In a preclinical study on choline supplementation, there was significant improvement in the spatial memory performance of healthy normal rats treated with choline at embryonic ages of 12–17 days and 16–30 days. Dendritic spine branching and spread was higher in CA1 pyramidal and upper and lower limb dorsal ganglion granule cells from control. Supplementation was also correlated with a higher amount of acetylcholine in hippocampal regions as opposed to the control supporting good cognition over time [52]. The potential biochemical pathways driving related behavioral changes such as improved spatial and cued navigational abilities was investigated after choline was supplemented [53]. They found that choline enhanced the phosphorylation of hippocampal MAPK and CREB signaling pathways, whereas deficiency of choline reduced it. Lastly, another perspective was explored on prenatal choline’s effect on risk or resilience behavior in the cognition and preservation of hippocampal plasticity in old age [54]. Researchers found that prenatal exposure only attenuated but did not prevent age-related decline in risky behavior. Additionally, these male and female rats showed some preserved hippocampal plasticity with age, indicating some protective effect [54].

Furthermore, looking at diseased models, choline supplementation reduced disease markers in Alzheimer’s and hippocampal-dependent memory impairment. In the Alzheimer’s model, it was reported that supplementation with choline significantly decreased amyloid-plaque load and improved spatial memory in the APP/PS1 Alzheimer’s disease mouse model [51]. The only limitation of this study is that only female mice were studied, so gender-dependent variations could occur and need to be further studied. Similarly, long-term dietary supplementation of CDP-choline could improve hippocampal-dependent memory impairment in rat models raised in food-impoverished conditions versus the control [53]. However, there was no improvement seen in food-enriched rats between the control and CDP-choline supplemented.

In addition to preclinical studies on choline’s benefits in normal aging and disease models, there were also studies showing other neuroprotective effects of choline in the brain and its enhancement of sensory modality processing. A study investigated prenatal choline supplementation and nerve growth factor (NGF) levels in the hippocampal and frontal cortex region. Researchers reported that there was significantly increased NGF in choline-supplemented rats. In the 20- and 90-day-old rats, there was a 25–30% NGF increase in the hippocampal region. However, in the frontal cortex, there was only a 16-fold increase in the 20-day-old rats, whereas there was a decrease in NGF in the 90-day-old rats [55]. In relation to the timing of auditory and visual processing, researchers found that there were differences in temporal integration between auditory and visual stimuli in aged rats when supplemented with prenatal choline. The auditory signal had faster duration discrimination than visual stimuli in choline-supplemented rats. They also showed increased attention and memory during adulthood with reduced age-related decline in cognition [56].

## 6. Future Directions and Conclusions

As humans age, the ability to absorb nutrients from our diet decreases. Reduced levels of 1C can impact brain health by increasing the risk for diseases and worsen outcomes when the brain is stressed by disease. As our review article has demonstrated, supplementation with folic acid, vitamin B12, or choline can have positive effects on normal and diseased brain function. 

The clinical trials reviewed in this study lack longevity amongst trials in the elderly population. Rarely do studies surpass 24 months, let alone 5 or 10 years. It also remains increasingly difficult to maintain a properly powered trial due to exclusion criteria like loss of life or failure to comply with the treatment regimen. Additional research needs to be conducted with more participants, including comparable participation amongst both males and females, and a control group properly age matched. Trials that follow these improvements will provide stronger-powered data to contribute to this critical research.

Maintaining adequate levels of 1C can promote healthy aging; this can take place through dietary supplementation. We propose the use of precision medicine to guide healthcare providers in implementing ideal supplementation for patients. This could include genetic testing for polymorphisms of enzymes involved in 1C or blood tests measuring levels of 1C metabolites [57,58] and then customizing supplementation to specific needs. Over-supplementation of 1C, specifically folic acid, has recently become a concern in childbearing women and the elderly. This is something that should be avoided, as the data do not show any positive health outcomes. 

Combination therapies have proven to be effective for patients suffering from ischemic stroke [59] as well as other neurological diseases [60]. We propose that nutrition should be added to a therapeutic plan for patients in addition to other interventions. The timing of dietary supplementation with 1C may be something that requires further investigation, since the onset of neurodegeneration is thought to occur well before the onset of symptoms [61,62,63]. 

Healthy aging is possible. In this review article, we propose that nutrition, especially 1C, plays an important role in promoting healthy neurological function through the aging process.

## Figures and Tables

**Figure 1 nutrients-15-03891-f001:**
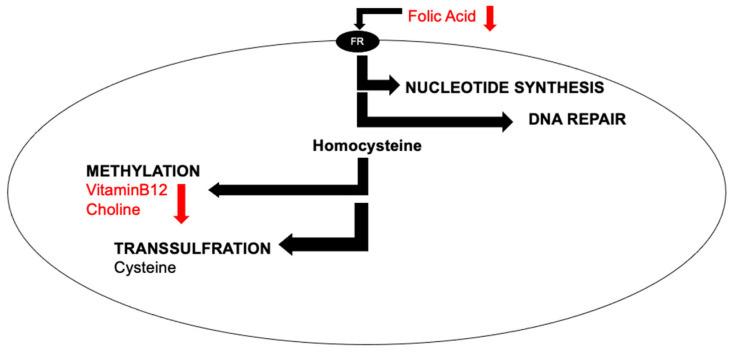
Simplified cellular one-carbon (1C) metabolism. B-vitamins are pleiotropic molecules, as they are involved in nucleotide synthesis, DNA repair, methylation, and transsulfuration. In this review, we focus on the impact of increasing dietary levels of folic acid, vitamin B12, and choline (red text). A control diet with adequate levels will also be used. Abbreviations: folate receptor, FR.

## Data Availability

Not applicable.

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
