# Peer review of "The Role of One-Carbon Metabolism in Healthy Brain Aging"

_nutrients, 2023, doi:10.3390/nu15183891_

Round 1

Reviewer 1 Report

Summary. In the manuscript entitled "The role of one-carbon metabolism on healthy brain aging," the authors summarize the current clinical and preclinical studies investigating dietary supplementation of 1C, specifically folic acid, choline, and vitamin B12 on healthy aging. This article also discusses future steps to consider for supplementation.

Overall, this is a valuable review. Nevertheless, in its current form, this article has several issues that should be carefully addressed.

Critiques:

General:

1.            Scientific language needs careful professional editing.

2.            It is a regular review article (not a comprehensive review); therefore, it should not include the material and methods sections and the results as well.

3.            Some references do not correspond with the description. Please double-check.

4.            The article contains descriptions of several animal studies that are not related to the subject chosen by the authors. For example, the authors review several studies done in juvenile animals or even pups, even though their topic is healthy brain aging. Sometimes, there are no references for states.

a.       Page 4. The authors state that "There is also evidence that folate supplementation in individuals with vitamin B12 deficiency can result in cognitive impairment, although the mechanism behind this is unknown." However, they do not provide a reference.

b.      Page 4. The same problem with the statement, "Research also suggests that in addition to decreased levels of B12 in healthy individuals, the brain atrophies at a slow, steady rate, decreasing in volume over time; but is exacerbated when deficient in vitamin B12."

c.        Page 6. The authors state that "ss the brain ages, total volume decreases, as well as glucose metabolism in the brain. Folic acid supplementation can prevent cognitive impairment, but its role in glucose metabolism is unknown" and provide reference #41. However, the provided reference is on choline sensitivity. Moreover, the study was done in rats. Please provide a relevant reference.

d.      Page 6. The authors mention "a study aimed to identify if folic acid status resulted in structural or metabolic changes in the brain. Three-month-old male rats were administered diets containing either 0.1 (folic acid deficient), 2.0 (folic acid normal), 4.0 (low folic acid supplementation), or 8.0mg (high folic acid supplementation) per kg diet for 22 months. Magnetic resonance imaging (MRI) and diffusion tensor imaging was completed to determine changes in brain structure." There is no relevant reference here as well.

e.       Page 7. The authors mention "a preclinical study on choline supplementation" with reference #40. However, this study was done in embryonic rats aged of 12-17 days and 16-30 days and is not relevant to the subject of the review.

5.            There are differences in the font chosen on pages 4 and 5.

6.            The first paragraph on page 6, with references 39, and 40, is unrelated to brain aging and should be excluded.

7.            Reference #54 is a self-citation and is not relevant to the statement.

Summary. In the manuscript entitled "The role of one-carbon metabolism on healthy brain aging," the authors summarize the current clinical and preclinical studies investigating dietary supplementation of 1C, specifically folic acid, choline, and vitamin B12 on healthy aging. This article also discusses future steps to consider for supplementation.

Overall, this is a valuable review. Nevertheless, in its current form, this article has several issues that should be carefully addressed.

Critiques:

General:

1.            Scientific language needs careful professional editing.

2.            It is a regular review article (not a comprehensive review); therefore, it should not include the material and methods sections and the results as well.

3.            Some references do not correspond with the description. Please double-check.

4.            The article contains descriptions of several animal studies that are not related to the subject chosen by the authors. For example, the authors review several studies done in juvenile animals or even pups, even though their topic is healthy brain aging. Sometimes, there are no references for states.

a.       Page 4. The authors state that "There is also evidence that folate supplementation in individuals with vitamin B12 deficiency can result in cognitive impairment, although the mechanism behind this is unknown." However, they do not provide a reference.

b.      Page 4. The same problem with the statement, "Research also suggests that in addition to decreased levels of B12 in healthy individuals, the brain atrophies at a slow, steady rate, decreasing in volume over time; but is exacerbated when deficient in vitamin B12."

c.        Page 6. The authors state that "ss the brain ages, total volume decreases, as well as glucose metabolism in the brain. Folic acid supplementation can prevent cognitive impairment, but its role in glucose metabolism is unknown" and provide reference #41. However, the provided reference is on choline sensitivity. Moreover, the study was done in rats. Please provide a relevant reference.

d.      Page 6. The authors mention "a study aimed to identify if folic acid status resulted in structural or metabolic changes in the brain. Three-month-old male rats were administered diets containing either 0.1 (folic acid deficient), 2.0 (folic acid normal), 4.0 (low folic acid supplementation), or 8.0mg (high folic acid supplementation) per kg diet for 22 months. Magnetic resonance imaging (MRI) and diffusion tensor imaging was completed to determine changes in brain structure." There is no relevant reference here as well.

e.       Page 7. The authors mention "a preclinical study on choline supplementation" with reference #40. However, this study was done in embryonic rats aged of 12-17 days and 16-30 days and is not relevant to the subject of the review.

5.            There are differences in the font chosen on pages 4 and 5.

6.            The first paragraph on page 6, with references 39, and 40, is unrelated to brain aging and should be excluded.

7.            Reference #54 is a self-citation and is not relevant to the statement.

Author Response

Reviewer 1

Summary. In the manuscript entitled "The role of one-carbon metabolism on healthy brain aging," the authors summarize the current clinical and preclinical studies investigating dietary supplementation of 1C, specifically folic acid, choline, and vitamin B12 on healthy aging. This article also discusses future steps to consider for supplementation.

Overall, this is a valuable review. Nevertheless, in its current form, this article has several issues that should be carefully addressed.

Response: We thank the reviewer for these comments and for helping us enhance our manuscript.

Critiques:

General:

  1. Scientific language needs careful professional editing.

Response: We apologize for this oversight and have gone through our paper in detail to rectify any scientific language issues.

  1. It is a regular review article (not a comprehensive review); therefore, it should not include the material and methods sections and the results as well.

Response: We have removed the materials and methods, as well as the result sections from the manuscript text.

  1. Some references do not correspond with the description. Please double-check.

Response: We apologize for this error; we have gone through the paper and double checked the references.

  1. The article contains descriptions of several animal studies that are not related to the subject chosen by the authors. For example, the authors review several studies done in juvenile animals or even pups, even though their topic is healthy brain aging. Sometimes, there are no references for states.
  2. Page 4. The authors state that "There is also evidence that folate supplementation in individuals with vitamin B12 deficiency can result in cognitive impairment, although the mechanism behind this is unknown." However, they do not provide a reference.

Response: We apologize for this oversight and have provided a reference.

Revised sentence: There is also evidence that folate supplementation in individuals with vitamin B12 deficiency can result in cognitive impairment [25].

[25] Christen, W.G.; Glynn, R.J.; Chew, E.Y.; Albert, C.M.; Manson, J.E. Folic Acid, Pyridoxine, and Cyanocobalamin Combination Treatment and Age-Related Macular Degeneration in Women: The Women’s Antioxidant and Folic Acid Cardiovascular Study. Arch. Intern. Med. 2009, 169, 335–341, doi:10.1001/archinternmed.2008.574.

  1. Page 4. The same problem with the statement, "Research also suggests that in addition to decreased levels of B12 in healthy individuals, the brain atrophies at a slow, steady rate, decreasing in volume over time; but is exacerbated when deficient in vitamin B12."

Response: We apologize for this oversight and have provided a reference.

Revised sentence: Research also suggests that in addition to decreased levels of B12 in healthy individuals, the brain atrophies at a slow, steady rate, decreasing in volume over time; but is exacerbated when deficient in vitamin B12[30].

[30] Douaud, G.; Refsum, H.; de Jager, C. a; Jacoby, R.; Nichols, T.E.; Smith, S.M.; Smith,  a D. Preventing Alzheimer’s Disease-Related Gray Matter Atrophy by B-Vitamin Treatment. Proc. Natl. Acad. Sci. U. S. A. 2013, 110, 9523–9528, doi:10.1073/pnas.1301816110.

  1. Page 6. The authors state that "ss the brain ages, total volume decreases, as well as glucose metabolism in the brain. Folic acid supplementation can prevent cognitive impairment, but its role in glucose metabolism is unknown" and provide reference #41. However, the provided reference is on choline sensitivity. Moreover, the study was done in rats. Please provide a relevant reference.

Response: We apologize for this oversight and have provided a reference.

Revised sentence: As the brain ages, total volume decreases [46], as well as glucose metabolism [47]  in the brain. Folic acid supplementation can prevent cognitive impairment, but its role in glucose metabolism is unknown.

[46] Lippelt, D.P.; Kint, S. van der; Herk, K. van; Naber, M. No Acute Effects of Choline Bitartrate Food Supplements on Memory in Healthy, Young, Human Adults. PLOS ONE 2016, 11, e0157714, doi:10.1371/journal.pone.0157714.

[47] Fotenos, A.F.; Snyder, A.Z.; Girton, L.E.; Morris, J.C.; Buckner, R.L. Normative Estimates of Cross-Sectional and Longitudinal Brain Volume Decline in Aging and AD. Neurology 2005, 64, 1032–1039, doi:10.1212/01.WNL.0000154530.72969.11.

  1. Page 6. The authors mention "a study aimed to identify if folic acid status resulted in structural or metabolic changes in the brain. Three-month-old male rats were administered diets containing either 0.1 (folic acid deficient), 2.0 (folic acid normal), 4.0 (low folic acid supplementation), or 8.0mg (high folic acid supplementation) per kg diet for 22 months. Magnetic resonance imaging (MRI) and diffusion tensor imaging was completed to determine changes in brain structure." There is no relevant reference here as well.

Response: We apologize for this oversight and have provided a reference.

Revised sentence: Researchers  aimed to identify if folic acid status resulted in structural or metabolic changes in the brain [48].

[48] Deery, H.A.; Di Paolo, R.; Moran, C.; Egan, G.F.; Jamadar, S.D. Lower Brain Glucose Metabolism in Normal Ageing Is Predominantly Frontal and Temporal: A Systematic Review and Pooled Effect Size and Activation Likelihood Estimates Meta‐analyses. Hum. Brain Mapp. 2022, 44, 1251–1277, doi:10.1002/hbm.26119.

  1. Page 7. The authors mention "a preclinical study on choline supplementation" with reference #40. However, this study was done in embryonic rats aged of 12-17 days and 16-30 days and is not relevant to the subject of the review.

Response: We thank the reviewer for bringing this to our attention. We do think this study adds to this review in terms of understanding how choline supplementation changes the brain. We have added the following statements to the manuscript text.

We have also included a preclinical study section that provides details on the mechanisms through which supplementation maybe changing brain function to promote healthy aging.

It is important to note that some preclinical studies we have included in this review article were not conducted in aged model systems. There are several challenges involved when using aged animals such as increased costs and translation to humans [44,45].

  1. There are differences in the font chosen on pages 4 and 5.

Response: We apologize for this and have made the font the same throughout the length of the manuscript.

  1. The first paragraph on page 6, with references 39, and 40, is unrelated to brain aging and should be excluded.

Response: This paragraph has been removed.

  1. Reference #54 is a self-citation and is not relevant to the statement.

            Response: The reference is a self-citation that describes how neurodegeneration occurs earlier that symptom onset, we have added 2 other citations for a different research groups.

The timing of dietary supplementation with 1C may be something that requires further investigation, since the onset of neurodegeneration is thought to occur well before the onset of symptoms [62–64]. 

[62]     Mau, K.J.; Jadavji, N.M. A New Perspective on Parkinson’s Disease: Pathology Begins in the Gastrointestinal Tract. J. Young Investig. 2017, 33, 1–8, doi:10.22186/jyi.33.4.63-70.

[63]     Hörder, H.; Johansson, L.; Guo, X.; Grimby, G.; Kern, S.; Östling, S.; Skoog, I. Midlife Cardiovascular Fitness and Dementia: A 44-Year Longitudinal Population Study in Women. Neurology 2018, 90, e1298–e1305, doi:10.1212/WNL.0000000000005290.

[64]     Suvila, K.; Lima, J.A.C.; Yano, Y.; Tan, Z.S.; Cheng, S.; Niiranen, T.J. Early-but Not Late-Onset Hypertension Is Related to Midlife Cognitive Function. Hypertension 2021, 77, 972–979, doi:10.1161/HYPERTENSIONAHA.120.16556.

Reviewer 2 Report

The authors conducted a comprehensive review encompassing both clinical and preclinical studies that investigate the effects of dietary supplementation of 1-carbon (1C) compounds, including folic acid, choline, and vitamin B12, on the process of healthy aging. The update was conducted meticulously, addressing the core aspects highlighted by the authors. However, a few minor concerns that warrant attention, outlined as follows:

1, Kindly consider removing the "Materials and Methods" section, while also revising the title of the "Results" section.

2, In order to enhance reader accessibility, it is suggested to break down the lengthy "3.1 Folic Acid" section into more discrete sub-sections, such as "Functional Role of Folic Acid," "Folic Acid's Implications in Alzheimer's Disease," and "Folic Acid Interactions with Vitamin B12."

3, It is pertinent to include Vitamin B9 (folate) alongside Vitamin B12 within the section dedicated to the role of essential water-soluble vitamins in 1-carbon metabolism.

4, To enrich the content, it is recommended to establish a dedicated section that delves into the associations between 1-carbon metabolism and age-related disorders, such as Alzheimer's disease, Parkinson's disease, and age-related macular degeneration (AMD).

Author Response

Reviewer 2

The authors conducted a comprehensive review encompassing both clinical and preclinical studies that investigate the effects of dietary supplementation of 1-carbon (1C) compounds, including folic acid, choline, and vitamin B12, on the process of healthy aging. The update was conducted meticulously, addressing the core aspects highlighted by the authors. However, a few minor concerns that warrant attention, outlined as follows:

Response: We thank the reviewer for their feedback.

1, Kindly consider removing the "Materials and Methods" section, while also revising the title of the "Results" section.

Response: We have removed the materials and methods, as well as the result sections from the manuscript text.

2, In order to enhance reader accessibility, it is suggested to break down the lengthy "3.1 Folic Acid" section into more discrete sub-sections, such as "Functional Role of Folic Acid," "Folic Acid's Implications in Alzheimer's Disease," and "Folic Acid Interactions with Vitamin B12."

Response: Thank-you for this suggestion, we have broken down the section. We have copied the revised text below.

2.1.  Functional Role of Folic Acid

Homocysteine plays an important role in methylation reactions in the body and is dependent on folic acid, as well as vitamins B6 and B12, for synthesis [13]. Folic acid deficiency results in elevated homocysteine levels and hyperhomocysteinemia has been identified as a risk factor for various diseases, such as vascular disease. In patients with hypertension the role of folic acid supplementation on SAH (S-adenosylhomocysteine) levels was measured [14]. This study also investigated the effect of the 5,10-methylenetetrahydrofolate reductase (MTHFR) C677T gene polymorphism on SAH levels. MTHFR is another enzyme that plays a role in folate metabolism. The patients recruited for this study were ages 45-75 years old with a history of primary hypertension and were grouped based on their MTHFR C677T polymorphism (CC, CT, or TT). The treatment groups included a daily oral dose of 10mg enalapril with either 0.4, 0.6, 0.8, 1.2, 1.6, 2.0 or 2.4mg of folic acid. Treatment groups administered 10mg enalapril with 0.4-2.0mg folic acid did not have altered S-adenosylhomocysteine (SAH) levels, but supplementation with 2.4mg increased SAH levels. Patients with the MTHFR C677T genotype CT and TT supplemented with 2.4mg folic acid had the increased SAH levels. The CC genotype did not have an increase in SAH levels with supplementation of 2.4mg folic acid. These findings indicated that the MTHFR genotypes affected by folic acid supplementation were both CT and TT. Additionally, higher levels of folic acid supplementation (2.4mg) resulted in increased homocysteine levels like what is reported during folate deficiency. This suggests that higher levels of folic acid supplementation could be harmful, which was confirmed by two other studies [15,16]. Low and high serum folate concentrations were associated with increased risk of mortality related to cardiovascular disease.

Telomere attrition or shortening is a key finding in many age-related disorders. The role of folic acid in telomere shortening was investigated, noting that folic acid had been implicated in astrocyte (glial cell) apoptosis and in aging disorders, but its role in telomere attrition was unknown [17]. Four-month-old male mice were fed diets with varying folic acid concentrations, including 0.1 (folic acid deficient diet), 2.0 (folic acid normal diet), 2.5 (low folic acid supplemented diet), and 3.0mg (high folic acid supplemented) of folic acid per kg of diet for six months. The mice were euthanized when they were 10 months old and astrocytes in the hippocampal and cerebral cortex tissue were analyzed, along with telomere length and telomerase activity. Folic acid supplementation prevented astrocyte apoptosis, telomere shortening and apoptosis, and degeneration in both the hippocampus and cortex. Telomerase activity was also shown to increase with folic acid supplementation, which is likely what prevented telomere attrition.

2.2.  Folic Acid's Implications in Neurodegeneration

The role of 1C metabolism in cognitive health has been explored [18,19]. Altered 1C metabolism through dietary deficiencies in B-vitamins may have a role in cognitive decline resulting from changes in DNA methylation [18], increased homocysteine levels [19] and decreased levels of neurotransmitters, as well as nucleotides [20]. Inflammation has been implicated in cognitive impairment and conditions such as Alzheimer’s disease [21]. The link between inflammation and folate status is also being studied, specifically the role of folic acid supplementation on inflammation and cognition in patients with Alzheimer’s disease. The participants for this study were patients aged 40-90 years old with new diagnoses of mild to moderate Alzheimer’s disease who were being treated with Donepezil. The control group included patients taking just the Donepezil and the treatment group included patients taking Donepezil with 1.25mg oral folic acid daily for six months with assessment every six weeks. Inflammation levels were determined through measurement of IL-6 and TNF? mRNA. The effect of folic acid supplementation on present Alzheimer’s disease pathology was determined through measurements of serum Aβ (amyloid beta) 40, Aβ 42, Aβ 42/Aβ 40, APP (amyloid precursor protein) mRNA, PS1 (presenilin 1) mRNA and PS2 mRNA. Cognition was studied through Mini-Mental State Examination (MMSE) score and Activities of Daily Living (ADL) score. Participants in the treatment group had higher levels in the MMSE compared to the control group indicating improved cognition with folate supplementation. Although, levels of A? 40 decreased after folic acid supplementation leading to a higher Aβ 42/Aβ 40 ratio which is typically the case in familial Alzheimer’s disease. SAM (S-adenosylmethionine) increased post-treatment and TNF? mRNA decreased indicating lower levels of inflammation. Prior studies have demonstrated a relationship between increasing SAM levels and decreasing TNF?.  It was concluded that folic acid supplementation along with Donepezil treatment improved cognition, as well as inflammation in patients with Alzheimer’s disease.

Another study investigated folic acid and vitamin B12 supplementation on inflammation and cognitive health in Alzheimer’s disease patients [22]. The patients for this study had a diagnosis of “stable” Alzheimer’s disease with a Montreal Cognitive Assessment score of less than 22 and were taking medication individually prescribed to them for their diagnosis. The treatment group was administered 1.2mg of folic acid and 50μg vitamin B12 orally per day for six months, while the control or placebo group was administered the equivalent number of starch tablets resembling both folic acid and vitamin B12. Blood samples were analyzed for folate, vitamin B12, SAM and SAH levels. There was also quantification of Aβ 40, Aβ 42, and the following inflammatory markers: IL2, IL6, IL10, MCP1, and TNF-?. Cognitive health was tested through neuropsychological testing including the Montreal Cognitive Assessment and the Alzheimer’s Disease Assessment Scale-Cognitive subscale. Supplementation of both folic acid and vitamin B12 resulted in higher total, naming, and orientation scores of the Montreal Cognitive Assessment. Scoring for attention improved for the Alzheimer’s Disease Assessment Scale-Cognitive subscale. The results of this neuropsychiatric testing indicated that the combined supplementation improved cognition in these patients. Combined supplementation also increased SAM and SAM/SAH levels and decreased homocysteine levels. Levels of inflammatory marker TNF? also decreased indicating a decrease in inflammation with combined folic acid and vitamin B12 supplementation.

A similar study aimed to determine the effect of folic acid and vitamin B12 (separately and combined) on patients with mild cognitive impairment (MCI) [9].  Participants were divided into four groups, including the control group that was not administered treatment, folic acid alone with 400μg of folic acid daily, vitamin B12 alone with 25μg daily, and combined treatment with 400μg folic acid and 25μg vitamin B12 daily for 6 months. Blood analysis was completed for quantification of inflammatory cytokines: IL2, IL6, IL10, TNFα, IFN-y and MCP-1. Cognitive testing was completed through the Wechsler Adult Intelligence Scale-Revised (WAIS-RC). Findings revealed that combined treatment decreased inflammatory cytokines IL6, TNFα, and MCP-1 while improving cognitive testing (WAIS-RC) scores.

Another example of the positive effects of dietary supplementation with folic acid, vitamin B6, vitamin B12 have been reported to reduce the risk of age-related macular degeneration (AMD), which is the leading cause of severe irreversible vision loss in the elderly [23]. The study supplemented female participant that had a preexisting cardiovascular disease with 2.5 mg/d, vitamin B6 50mg/d, and vitamin B12 1mg/d for 7.3 years, there were 55 cases of AMD in he treated group and 82 in the placebo group. Other studies investigating 1C supplementation have also reported that dietary supplementation is beneficial for people at high risk of AMD [24,25].

2.3. Folic Acid Interactions with Vitamin B12

The effect of high serum folate was studied in elderly individuals with a history of diabetes and vitamin B12 deficiency [26]. Prior studies have shown that high folate supplementation can result in increased cognitive impairment [27]. There is also evidence that folate supplementation in individuals with vitamin B12 deficiency can result in cognitive impairment [28]. The goal of this study was to determine if there was a correlation between high levels of folate supplementation and cognitive impairment in individuals with vitamin B12 deficiency. The participants for this study were recruited from an established study of vitamin B12 supplementation in elderly Chinese individuals (average of 75 years old) with a history of diabetes. Serum methylmalonic acid and folate levels were assessed in these individuals and those with high methylmalonic acid with a concentration greater than 0.3μmol and high serum folate with a concentration greater than 31.4nmol/L were chosen for this study. Magnetic resonance imaging was used to analyze brain structure in relation to methylmalonic and folate levels in these individual. Imaging revealed atrophy of the gray matter in the right middle occipital gyrus, as well as the inferior frontal gyrus, in individuals with high folate concentrations and vitamin B12 deficiency. This led to the conclusion that high folate concentrations could be harmful to neuronal structure resulting in degeneration in the setting of vitamin B12 deficiency.

3, It is pertinent to include Vitamin B9 (folate) alongside Vitamin B12 within the section dedicated to the role of essential water-soluble vitamins in 1-carbon metabolism.

Response: We thank the reviewer for this comment. We would like to note that there is a section in the revised manuscript linking both folic acid and vitamin B12 together, thus emphasizing the of essential water-soluble vitamins in one-carbon metabolism. We have copied the section from the test below.

2.3. Folic Acid Interactions with Vitamin B12

The effect of high serum folate was studied in elderly individuals with a history of diabetes and vitamin B12 deficiency [26]. Prior studies have shown that high folate supplementation can result in increased cognitive impairment [27]. There is also evidence that folate supplementation in individuals with vitamin B12 deficiency can result in cognitive impairment [28]. The goal of this study was to determine if there was a correlation between high levels of folate supplementation and cognitive impairment in individuals with vitamin B12 deficiency. The participants for this study were recruited from an established study of vitamin B12 supplementation in elderly Chinese individuals (average of 75 years old) with a history of diabetes. Serum methylmalonic acid and folate levels were assessed in these individuals and those with high methylmalonic acid with a concentration greater than 0.3μmol and high serum folate with a concentration greater than 31.4nmol/L were chosen for this study. Magnetic resonance imaging was used to analyze brain structure in relation to methylmalonic and folate levels in these individual. Imaging revealed atrophy of the gray matter in the right middle occipital gyrus, as well as the inferior frontal gyrus, in individuals with high folate concentrations and vitamin B12 deficiency. This led to the conclusion that high folate concentrations could be harmful to neuronal structure resulting in degeneration in the setting of vitamin B12 deficiency.

4, To enrich the content, it is recommended to establish a dedicated section that delves into the associations between 1-carbon metabolism and age-related disorders, such as Alzheimer's disease, Parkinson's disease, and age-related macular degeneration (AMD).

Response: We thanks the reviewer for this suggestion. We have added more text to the manuscript on AMD, below is a copy of the additional text. This text has been added to section titled, 2.2.  Folic Acid's Implications in Neurodegeneration

Another example of the positive effects of dietary supplementation with folic acid, vitamin B6, vitamin B12 have been reported to reduce the risk of age-related macular degeneration (AMD), which is the leading cause of severe irreversible vision loss in the elderly [23]. The study supplemented female participant that had a preexisting cardiovascular disease with 2.5 mg/d, vitamin B6 50mg/d, and vitamin B12 1mg/d for 7.3 years, there were 55 cases of AMD in he treated group and 82 in the placebo group. Other studies investigating 1C supplementation have also reported that dietary supplementation is beneficial for people at high risk of AMD [24,25].